# Sodium–Glucose Cotransporter 2 Inhibitor Combined with Conventional Diuretics Ameliorate Body Fluid Retention without Excessive Plasma Volume Reduction

**DOI:** 10.3390/diagnostics14111194

**Published:** 2024-06-05

**Authors:** Maki Asakura-Kinoshita, Takahiro Masuda, Kentaro Oka, Ken Ohara, Marina Miura, Masato Morinari, Kyohei Misawa, Yasuharu Miyazawa, Tetsu Akimoto, Kazuyuki Shimada, Daisuke Nagata

**Affiliations:** 1Division of Nephrology, Department of Internal Medicine, Jichi Medical University, Shimotsuke 3290498, Japan; two-away-zone@hotmail.com (M.A.-K.); ninetofiveman.ken@gmail.com (K.O.); ko0235082@gmail.com (K.O.); r1928mk@jichi.ac.jp (K.M.); tetsu-a@jichi.ac.jp (T.A.); ngtdsktky@gmail.com (D.N.); 2Department of Nephrology, Shin-Oyama City Hospital, Oyama 3230827, Japan; wildcat.strawberry@gmail.com; 3Department of Internal Medicine, Nasu Minami Hospital, Nasu-Karasuyama 3210621, Japan; m.morinari@gmail.com (M.M.); hal@dd.iij4u.or.jp (Y.M.); 4Department of Cardiology, Shin-Oyama City Hospital, Oyama 3230827, Japan; kazuyuki@hospital.oyama.tochigi.jp

**Keywords:** sodium–glucose cotransporter 2 (SGLT2) inhibitor, copeptin, vasopressin, plasma volume, interstitial fluid, extracellular water, cardiorenal protection, fluid retention, loop diuretic, heart failure

## Abstract

We previously reported that sodium–glucose cotransporter 2 (SGLT2) inhibitors exert sustained fluid homeostatic actions through compensatory increases in osmotic diuresis-induced vasopressin secretion and fluid intake. However, SGLT2 inhibitors alone do not produce durable amelioration of fluid retention. In this study, we examined the comparative effects of the SGLT2 inhibitor dapagliflozin (SGLT2i group, *n* = 53) and the combined use of dapagliflozin and conventional diuretics, including loop diuretics and/or thiazides (SGLT2i + diuretic group, *n* = 23), on serum copeptin, a stable, sensitive, and simple surrogate marker of vasopressin release and body fluid status. After six months of treatment, the change in copeptin was significantly lower in the SGLT2i + diuretic group than in the SGLT2i group (−1.4 ± 31.5% vs. 31.5 ± 56.3%, *p* = 0.0153). The change in the estimated plasma volume calculated using the Strauss formula was not significantly different between the two groups. Contrastingly, changes in interstitial fluid, extracellular water, intracellular water, and total body water were significantly lower in the SGLT2i + diuretic group than in the SGLT2i group. Changes in renin, aldosterone, and absolute epinephrine levels were not significantly different between the two groups. In conclusion, the combined use of the SGLT2 inhibitor dapagliflozin and conventional diuretics inhibited the increase in copeptin levels and remarkably ameliorated fluid retention without excessively reducing plasma volume and activating the renin–angiotensin–aldosterone and sympathetic nervous systems.

## 1. Introduction

Sodium–glucose cotransporter 2 (SGLT2) inhibitors suppress glucose and sodium reabsorption by inhibiting SGLT2 in the early proximal tubules [1]. In addition, because of the physical and functional coupling of SGLT2 to other transporters in the early proximal tubule, including NHE3 and URAT1, SGLT2 inhibitors reduce the reabsorption of not only glucose but also a larger amount of sodium than the expected effects of SGLT2 inhibition alone, thereby reducing volume retention [1,2,3,4,5,6,7,8]. In the first term from 2015 to 2019, large-scale randomized clinical trials in patients with diabetes revealed the cardiorenal protective effects of SGLT2 inhibitors [9,10,11,12]. Furthermore, in the second term of 2020, six clinical trials of patients with chronic kidney disease (CKD) or heart failure with or without diabetes showed that the SGLT2 inhibitors dapagliflozin and empagliflozin exhibited cardiorenal protective effects [13,14,15,16,17,18]. Based on these clinical trials, the use of SGLT2 inhibitors has expanded as a treatment for CKD and heart failure in patients with or without diabetes [19].

Various mechanisms, including the attenuation of glomerular hyperfiltration by a tubuloglomerular feedback mechanism, the increase in hemoglobin, and the elevation of ketone bodies, have been proposed to explain the cardiorenal protective mechanisms of SGLT2 inhibitors [1,2,20]. In addition, we have shown that SGLT2 inhibitors exert sustained fluid homeostatic actions, which may contribute to cardiorenal protection through the amelioration of the renin–angiotensin–aldosterone system (RAAS) and sympathetic nervous system (SNS) activation [4,5,19,21,22,23,24].

SGLT2 inhibitors cause mild natriuretic and glucosuria-induced osmotic diuresis by inhibiting SGLT2 in the early proximal tubules; however, they have a low risk of hypovolemia [5,21,22,24,25,26,27]. In our clinical studies, SGLT2 inhibitor dapagliflozin ameliorates fluid retention and maintains euvolemic status both in short-term and long-term treatment periods, suggesting a fluid homeostatic action by SGLT2 inhibitors [4,5,24]. The fluid homeostatic mechanism of SGLT2 inhibitors is due to (1) a compensatory increase in fluid intake in response to osmotic diuresis and (2) the suppression of excessive urine volume by vasopressin-induced solute-free water reabsorption in the renal collecting duct [21,22,23,28,29].

Conversely, clinical trials suggest that SGLT2 inhibitors do not produce a durable natriuresis or objective decongestion in patients with heart failure [30,31]. As osmotic diuresis induced by SGLT2 inhibitors evokes marked counter-regulatory activation of sodium and water reabsorption in the distal nephron segments [3,8,30,31,32,33], the single use of SGLT2 inhibitors without conventional diuretics may be insufficient for the optimization of body fluid status. We recently reported that some patients taking dapagliflozin without conventional diuretics had an increased extracellular fluid status for six months, whereas none of the patients taking dapagliflozin and conventional diuretics had an increased extracellular fluid status [4]. Furthermore, in euvolemic rats, the SGLT2 inhibitor ipragliflozin increased vasopressin-related stimulation of water reabsorption in the collecting duct and maintained body water, whereas the loop diuretic furosemide decreased vasopressin secretion and body water for one week [22].

Copeptin, the C-terminal moiety of provasopressin, is secreted by magnocellular hypothalamic neurons [34]. Vasopressin and copeptin are stimulated by similar physiological processes, such as osmotic stimulation, hypovolemia, or stress [35]. While vasopressin is difficult to measure due to complex pre-analytical requirements and technical reasons, copeptin has recently been used as a stable (several days at room temperature in serum or plasma), sensitive, and simple-to-measure surrogate vasopressin release marker [34,35].

Based on the aforementioned evidence, we hypothesized that the combined use of SGLT2 inhibitors and conventional diuretics, including loop diuretics and/or thiazides, suppresses vasopressin secretion and accelerates body fluid reduction compared to the single use of SGLT2 inhibitors. In this study, we examined the comparative effects of the SGLT2 inhibitor dapagliflozin alone and the combined use of dapagliflozin and conventional diuretics on the vasopressin surrogate marker copeptin and body fluid status. Dapagliflozin was selected among several SGLT2 inhibitors because it was the only SGLT2 inhibitor that was indicated for patients with CKD with or without diabetes during the entry period.

## 2. Materials and Methods

### 2.1. Patients

In this prospective, open-label, and non-randomized study, 101 patients with CKD at Shin-Oyama City Hospital (Oyama, Tochigi, Japan) and Nasu Minami Hospital (Nasukarasuyama, Tochigi, Japan) were treated with the SGLT2 inhibitor dapagliflozin between February 2016 and August 2022, as previously reported as the DAPA-BODY Trial [4]. During the entry period, 65 patients received dapagliflozin without diuretics (SGLT2i group), and 47 patients received dapagliflozin plus diuretics (SGLT2i + diuretic group). Patients in the SGLT2i + diuretic group were treated with dapagliflozin plus loop diuretics (furosemide, torsemide, or azosemide) and/or thiazides (trichlormethiazide or hydrochlorothiazide). In the SGLT2i group, 12 patients were excluded from the final analysis for the following reasons: one was lost to follow-up, eight had incomplete bioimpedance analysis (BIA) data, two discontinued dapagliflozin, and one died (heart failure). In the SGLT2i + diuretic group, 24 patients were excluded from the final analysis for the following reasons: two were lost to follow-up, 16 had incomplete BIA data, one started dialysis, three discontinued dapagliflozin, and two died (traffic accident and heart failure). Finally, 53 patients received dapagliflozin without conventional diuretics (dapagliflozin 5 mg, *n* = 3; 10 mg, *n* = 50) and 23 patients received dapagliflozin with conventional diuretics (dapagliflozin 5 mg, *n* = 1; 10 mg, *n* = 22; loop diuretics, 19 cases; thiazides, 3 cases; loop and thiazide diuretics, 1 case).

The decision to introduce dapagliflozin was made by the attending physicians based on the following criteria: (1) an estimated glomerular filtration rate (eGFR) between 15 mL/min/1.73 m^2^ and 59 mL/min/1.73 m^2^ with or without proteinuria; or (2) an eGFR of 60 mL/min/1.73 m^2^ or greater and positive proteinuria. Because dapagliflozin has been reported to induce favorable renal outcomes in patients with diabetes or chronic glomerular nephropathy [17,36], it has been introduced as the primary treatment for diabetes and chronic glomerular nephropathy. Diuretics were continued while patients received diuretics (loop or thiazide diuretics) at the time of dapagliflozin administration, with dose modifications by the attending physicians during the study period. The exclusion criteria were a history of renal replacement therapy, dialysis, type 1 diabetes, active malignancy, or pacemaker implantation.

This study was conducted in accordance with the ethical principles of the Declaration of Helsinki and was registered in the University Hospital Medical Information Network Clinical Trials Registration System (UMIN-CTR) as part of the DAPA-BODY Trial (UMIN000048568). The study protocol was approved by the independent ethics committees of Shin-Oyama City Hospital (approval number: SOR2020-004) and Nasu Minami Hospital (approval number: 2016-03). Written informed consent to participate in the study was obtained from all patients.

### 2.2. Study Endpoints

The primary endpoint of the study was body fluid status (estimated plasma volume [ePV], interstitial fluid [IF], extracellular water [ECW], intracellular water [ICW], total body water (TBW), and ECW/TBW) at six months. The secondary endpoints were changes in copeptin levels and the activities of the RAAS and SNS, including renin, aldosterone, and epinephrine.

### 2.3. Blood Analyses

Blood samples were collected at baseline (at the time of SGLT2 inhibitor administration) and six months later. The eGFR was calculated using the Modification of Diet in Renal Disease study coefficients modified for the Japanese population [37]. Serum copeptin, a surrogate marker for vasopressin, was assayed by Thermo Fisher Scientific using an automated immunoluminometric assay (ultra-sensitive B-R-A-H-M-S copeptin proAVP; Thermo Fisher Scientific, Hennigsdorf, Germany) [38]. Serum samples were also analyzed using enzyme-linked immunosorbent assay (ELISA) according to the manufacturer’s instructions: the renin ELISA kit (DRG Instruments GmbH, Frauenbergstr, Marburg, Germany) [4,39] and the aldosterone ELISA kit (IBL Co., Ltd., Fujioka, Japan) [4,40]. Plasma epinephrine was measured in the SRL laboratory (Hachioji, Tokyo, Japan) [4,41].

Changes in ePV were calculated using the Strauss formula as follows: hemoglobin baseline/hemoglobin six months × [(100 − hematocrit six months)/(100 − hematocrit baseline)-1] × 100 [22,42,43]. Absolute ePV was calculated using the Kaplan–Hakim formula: (1-hematocrit) × (a + [b × body weight (kg)], where a = 1530 in men and 864 in women, and b = 41 in men and 47.9 in women) [44,45,46].

### 2.4. Measurement of the Fluid Volume Using a BIA Device

Body fluid volume was assessed using a BIA device with eight tactile electrodes (InBody S10; InBody Japan, Tokyo, Japan) at the time of SGLT2 inhibitor administration and six months later, as in our previous studies [4,5,24,26,47]. ICW, ECW, TBW (ICW + ECW), and ECW/TBW were calculated from the sum of each segment using the equations in the BIA software program built into the InBody S10. Since ECW is the sum of IF and plasma volume, IF was calculated using the formula ECW—ePV (Kaplan–Hakim formula). ECW/TBW was used as a marker of fluid status as it is a marker of extracellular fluid status and a predictor of renal outcome [48].

### 2.5. Statistical Analyses

Data distributions were determined for normality using the Shapiro–Wilk test. Normally distributed data are represented as the mean and standard deviation. Data that were not normally distributed are presented as medians and interquartile ranges. Unpaired or paired *t*-tests were used to compare the results between the two groups, as appropriate. Correlations between variables were analyzed using Pearson’s correlation test. JMP 14.3.0 statistical software (SAS Institute, Inc., Cary, NC, USA) was used for statistical analysis, and statistical significance was set at *p* < 0.05.

## 3. Results

### 3.1. Patients’ Characteristics

Of the 76 enrolled patients with CKD, 53 and 23 were assigned to the SGLT2i and SGLT2i + diuretic groups, respectively. Hemoglobin, hematocrit, serum albumin, and eGFR were significantly higher in the SGLT2i group than in the SGLT2i + diuretic group (Table 1). Age, plasma glucose, brain natriuretic peptide (BNP), blood urea nitrogen (BUN), copeptin, renin levels, ECW/TBW, and prescription rates of loop diuretics, thiazide diuretics, and calcium channel blockers were significantly higher in the SGLT2i + diuretic group than in the SGLT2i group (Table 1).

### 3.2. Changes in Vasopressin Surrogate Marker Copeptin and Related Parameters

SGLT2 inhibitors significantly increased copeptin levels for 6 months (Appendix A). The change in copeptin from baseline to 6 months was significantly lower in the SGLT2i + diuretic group than in the SGLT2i group (SGLT2i group 31.5 ± 56.3% vs. SGLT2i + diuretic group −1.4 ± 31.5%, *p* = 0.0153) (Figure 1A). SGLT2 inhibitors combined with diuretics significantly decreased serum Na^+^ concentrations for six months (Appendix A). However, serum osmolality was not changed in the SGLT2i and SGLT2i + diuretic groups (Appendix A). Compared with the SGLT2i group, the SGLT2i + diuretic group exhibited significantly lower changes in serum Na^+^ concentration (−0.2 ± 1.4% vs. −1.0 ± 1.9%, *p* = 0.0245) (Figure 1B), while serum osmolality was not significantly different between the two groups (−0.05 ± 1.42% vs. −0.42 ± 2.06%, *p* = 0.2100) (Figure 1C).

### 3.3. Changes in Body Fluid Status

SGLT2 inhibitor alone and SGLT2 inhibitor combined with diuretics significantly decreased ePV and IF for six months (Appendix A, Table 2). The change in ePV was not significantly different between the SGLT2i group and the SGLT2i + diuretic group (−1.7 ± 5.2% vs. −2.6 ± 5.3%, *p* = 0.2565) (Figure 2A). In contrast, the change in IF was significantly lower in the SGLT2i + diuretic group than in the SGLT2i group (−1.7 ± 5.4% vs. −8.8 ± 12.5%, *p* = 0.0013) (Figure 2B). Consequently, the ΔIF-to-ΔePV ratio (ΔIF/ΔePV) was significantly higher in the SGLT2i + diuretic group than in the SGLT2i group (−1.1% ± 9.5% vs. 6.0% ± 23.1%, *p* = 0.0454) (Figure 2C).

SGLT2 inhibitor alone and SGLT2 inhibitor combined with diuretics significantly decreased ECW, ICW, TBW, and ECW/TBW for six months (Appendix A, Table 2). Compared with the SGLT2i group, the SGLT2i + diuretic group showed a significantly greater decrease in ECW (−2.2 ± 4.8% vs. −8.1 ± 10.1%, *p* = 0.0007), ICW (−2.1 ± 5.1% vs. −5.8 ± 9.3%, *p* = 0.0183), TBW (−2.1 ± 4.8% vs. −6.2 ± 9.0%, *p* = 0.0089) (Figure 3A–C), and ECW/TBW (−0.03 ± 1.73% vs. −2.06 ± 2.01%, *p* < 0.001) (Figure 3D).

The amount of fluid change after six months is shown in Table 2. Similar results were obtained, with no significant difference in the changes in ePV between the SGLT2i and SGLT2i + diuretic groups (−0.10 L vs. −0.16 L, *p* = 0.0908). In contrast, changes in IF (−0.19 ± 0.59 L vs. −1.02 ± 1.39 L, *p* = 0.0007), ECW (−0.23 ± 0.87 L vs. −1.16 ± 1.40 L, *p* = 0.0006), ICW (−0.39 ± 1.19 L vs. −1.03 ± 1.61 L, *p* = 0.0332), TBW (−0.62 ± 1.97 L vs. −2.05 ± 2.87 L, *p* = 0.0096), and ECW/TBW (0.001 ± 0.011 vs. −0.009 ± 0.009, *p* = 0.0003) were significantly and remarkably lower in the SGLT2i + diuretic group than in the SGLT2i group.

Baseline eGFR was not significantly correlated with the changes in copeptin (r = 0.209, *p* = 0.098), PV (r = −0.136, *p* = 0.258), IF (r = 0.059, *p* = 0.651), ECV (r = 0.067, *p* = 0.586), ICW (r = 0.048, *p* = 0.693), TBW (r = 0.039, *p* = 0.753), and ECW/TBW (r = 0.085, *p* = 0.480).

### 3.4. Changes in BNP and Parameters of RAAS and SNS

SGLT2 inhibitor alone and SGLT2 inhibitor combined with diuretics did not significantly change BNP, renin, or aldosterone for 6 months (Appendix A). Changes in BNP (26.3 ± 84.5% vs. 4.0 ± 60.8%, *p* = 0.1716), renin (16.3 ± 102.8% vs. 26.6 ± 91.0%, *p* = 0.3765), and aldosterone (9.8 ± 48.0% vs. 20.3 ± 51.98%, *p* = 0.2560) were similar between the SGLT2i group and the SGLT2i + diuretic group (Figure 4A–C). The adrenaline levels at 6 months were not significantly different between the SGLT2i and SGLT2i + diuretic groups (21.3 ± 13.8 pg/mL vs. 21.0 ± 10.4 pg/mL, *p* = 0.4854) (Figure 4C).

Taken together, the SGLT2i and SGLT2i + diuretic groups showed a decrease in body fluid volume; however, the SGLT2i + diuretic group showed a greater decrease in fluid volume without significant differences in the estimated decrease in circulating plasma volume, RAAS, or SNS activity (Figure 4 and Figure 5A,B).

## 4. Discussion

The combined use of the SGLT2 inhibitor dapagliflozin and conventional diuretics inhibited the increase in the vasopressin surrogate marker copeptin and markedly ameliorated fluid retention, including IF and ICW, whereas the SGLT2 inhibitor alone increased copeptin levels and mildly decreased body fluid volume. In contrast, the changes in ePV and parameters of RAAS and SNS activity did not change regardless of the use of conventional diuretics.

This study showed that serum copeptin, a surrogate biomarker of vasopressin, was not increased by the combined use of SGLT2 inhibitors and conventional diuretics. Copeptin, the C-terminal moiety of provasopressin, is secreted by magnocellular hypothalamic neurons in equimolar amounts in response to osmotic, hemodynamic, and stressful stimuli [35]. Copeptin can be measured in the plasma or serum and remains stable for several days [35]. Copeptin was strongly correlated with a wide range of osmolalities (Spearman’s rank correlation coefficient 0.8) [34]. Clinical studies in patients with and without diabetes have shown that the SGLT2 inhibitors dapagliflozin and empagliflozin increase copeptin levels to maintain euvolemic fluid status [28,49,50]. Furthermore, in diabetic and non-diabetic rats, the SGLT2 inhibitor ipragliflozin increased osmotic diuresis-induced vasopressin secretion and solute-free water reabsorption in the collecting duct, which contributed to maintaining a long-term euvolemic fluid status [21,22]. In sharp contrast to SGLT2 inhibitors, the loop diuretic furosemide decreased vasopressin secretion and solute-free water reabsorption, facilitating a negative fluid balance and reducing the ePV and body fluid volume [22]. However, the effects of the combined use of SGLT2 inhibitors and conventional diuretics on vasopressin secretion remain unevaluated. Therefore, to the best of our knowledge, this is the first study to show that an SGLT2 inhibitor plus conventional diuretics did not increase vasopressin secretion.

Why does the combined use of SGLT2 inhibitors and diuretics not increase copeptin levels? In our animal studies, the SGLT2 inhibitor ipragliflozin slightly increased serum Na^+^ concentration, though not serum osmolality, whereas the loop diuretic furosemide induced a small reduction in serum Na^+^ concentration [21,22]. As Na^+^ is an effective osmolyte for vasopressin release [51], the increase in serum Na^+^ concentrations induced by ipragliflozin might increase vasopressin release [22]. In the present study, the combined use of the SGLT2 inhibitor dapagliflozin and conventional diuretics (loop or thiazide diuretics) mildly but significantly decreased serum Na^+^ levels compared with the SGLT2 inhibitor alone. A possible reason for the reduction in serum Na^+^ levels in combination therapy is natriuresis caused by loop or thiazide diuretics. Therefore, the decrease in serum Na^+^ concentrations might result in a lack of an increase in copeptin levels with the combined use of SGLT2 inhibitors and conventional diuretics.

Notably, the degree of decrease in ePV was comparable between SGLT2 inhibitor alone and combination therapy with a conventional diuretic, while combination therapy markedly decreased IF and ICW compared to the single use of SGLT2 inhibitor. Similar to the present study, Shiina et al. reported that the SGLT2 inhibitor canagliflozin decreased ePV to a similar degree regardless of conventional diuretic use [52]. However, the effects of the combined use of SGLT2 inhibitors and conventional diuretics on fluid distribution remain unevaluated. Hallow et al. reported that the SGLT2 inhibitor dapagliflozin produced a 3-fold decrease in IF volume relative to blood volume, whereas the reduction in IF with the loop diuretic bumetanide was only 66% of the reduction in blood volume [53]. The present study showed that dapagliflozin alone produced about a 2-fold decrease in IF volume relative to blood volume (IF −0.19 vs. ePV −0.10 L). In contrast, the combination treatment of dapagliflozin and conventional diuretics (loop or thiazide diuretics) produced a 6-fold decrease in IF relative to blood volume (IF −1.02 vs. ePV, −0.16 L). In the comparison of the combination and single therapy of dapagliflozin, the changes in ePV were comparable (−0.16 L vs. −0.10 L), whereas the changes in IF in the combination therapy were more than 5-fold higher than that in dapagliflozin alone (−1.02 vs. −0.19 L). The changes in ICW in the combination therapy were 2.6-fold higher than those in dapagliflozin alone (−1.03 vs. −0.39 L). These data suggest that the combined use of SGLT2 inhibitors and conventional diuretics markedly ameliorates interstitial fluid retention without excessively reducing blood volume.

Why does the combined use of SGLT2 inhibitors and diuretics ameliorate interstitial fluid retention without excessive intravascular fluid loss? Three potential mechanisms are involved in this phenomenon. First, the initial transient decrease in circulating blood volume was due to an increase in urine output. The lack of an increase in copeptin levels in combination therapy may accelerate urine output by inhibiting vasopressin-induced water reabsorption in the collecting duct [21]. In a clinical trial in patients with type 2 diabetes and heart failure, the SGLT2 inhibitor empagliflozin, in addition to the loop diuretic furosemide, caused a significant increase in urine volume compared with furosemide alone [54]. Although the current study had no urine volume data, the larger increase in urine volume in the combination therapy group might have induced a transient decrease in the circulating blood volume. Second, an increase in the hydrostatic pressure gradient between the interstitial and intracapillary spaces was observed, which occurred due to the initial decrease in circulating blood volume. The decrease in intracapillary hydrostatic pressure due to accelerated diuresis in combination therapy had caused a negative transcapillary pressure gradient, which in turn induced a fluid shift from the interstitial to the intravascular compartment [47,55]. Third, an improvement in the increased vascular endothelial permeability induced by the SGLT2 inhibitor might accelerate the fluid shift from the interstitial into the intracellular space. SGLT2 inhibitors might limit inflammation by preserving the structural integrity of the vascular endothelial glycocalyx, which, in turn, limits vascular permeability [56]. In addition, a recent clinical study reported that SGLT2 inhibitors result in a significantly thicker endothelial glycocalyx than that observed in patients treated with insulin [57]. Therefore, in the current study, the amelioration of vascular permeability by the SGLT2 inhibitor dapagliflozin might have strengthened the fluid shift from the interstitial space into the intravascular space.

A remarkable reduction in interstitial fluid volume in combination therapy with SGLT2 inhibitors and conventional diuretics is useful for ameliorating interstitial fluid retention in patients with congestive heart failure, renal interstitial congestion, and ascites of liver cirrhosis. The combined use of SGLT2 inhibitors (dapagliflozin or empagliflozin) and conventional diuretics in patients with heart failure reduced the composite outcomes of cardiovascular death or worsening heart failure events [58,59]. These data suggest that combination therapy with SGLT2 inhibitors and conventional diuretics plays a crucial role in decongestion [60]. Furthermore, the SGLT2 inhibitor tofogliflozin suppressed renal interstitial fibrosis, inflammation, and mitochondrial dysfunction after renal venous congestion, especially in the renal cortex [61]. This result indicates that SGLT2 inhibitors are a therapeutic candidate for renal impairment associated with heart failure [61]. In another clinical setting, SGLT2 inhibitors were used to effectively treat ascites caused by liver cirrhosis [62,63]. Treatment with SGLT2 inhibitors, in addition to conventional diuretics, appears to be effective against furosemide- and spironolactone-resistant ascites [62,63].

Avoidance of excess plasma volume reduction in combination therapy with SGLT2 inhibitors and conventional diuretics may prevent the activation of the RAAS and SNS. The current study showed that renin, aldosterone, and epinephrine levels were similar regardless of conventional diuretic use, which may contribute to long-term cardiorenal protection even in patients receiving combination therapy [19] (Figure 5B). On the other hand, a subgroup analysis of the DAPA-HF trial and other real-world data showed that volume depletion events were significantly more common with dapagliflozin at higher-dose diuretics (furosemide or furosemide equivalent more than 40 mg) [64,65]. Therefore, lower-dose diuretics (furosemide equivalents less than 40 mg) may be an important safeguard against hypovolemia, which prevents the activation of the RAAS and SNS.

The various beneficial effects of SGLT2 inhibitors have been demonstrated in conjunction with ACE inhibitors, or ARBs [9,10,12]. In previous randomized clinical trials in patients with CKD (CREDENCE, DAPA-CKD, or EMPA-KIDNEY trials), the combined use of these drugs was over 80% [11,17,18], but it was approximately 60% in the current study. However, a relatively low rate of combination therapy might have weak effects on the difference in body fluid status between the two groups because (1) the primary effect of ACE inhibitors or ARBs is not diuretic action and (2) the usage rate of ACE inhibitors or ARBs was similar between the two groups.

The effect of SGLT2 inhibitors in the kidney depends on the degree of renal insufficiency. Therefore, the inhibition of glucose reabsorption in the proximal tubule is reduced with failing kidney function [66]. In this study, eGFR ranged roughly between 20 and 50 mL/min/1.73 m^2^, but baseline eGFR was not significantly correlated with the changes in body fluid status. Therefore, different baseline renal functions in enrolled patients might not strongly affect the main results.

This study had several limitations and remaining issues. First, this was not a randomized control study; therefore, several baseline characteristics differed between the two groups. These differences may potentially affect the results. Second, the number of enrolled patients was relatively small, similar to previous studies that focused on fluid status with SGLT2 inhibitors [28,67,68,69]. Third, studies on the use of conventional diuretics alone are lacking. Therefore, the comparative effect of the combined and single use of conventional diuretics remains uncertain. Fourth, comparative drug effects of potential natriuretics (loop diuretic, thiazide, and MR antagonist) are lacking because of a small number of drug usage (thiazide only, *n* = 2 and MR antagonist only, *n* = 2). Further studies are required to evaluate the difference between these drugs. Fifth, PV and changes in PV were estimated using the formulas (Kaplan–Hakim formula and Strauss formula). Although previous studies showed moderate to good correlations between measured PV using radioisotope assays and estimated PV [70], the accuracy of body fluid status, including PV and IF (ICW–PV), may have some data reliability limitations.

## 5. Conclusions

In conclusion, the combined use of the SGLT2 inhibitor dapagliflozin and conventional diuretics inhibited the increase in copeptin levels and markedly ameliorated fluid retention without excessively reducing plasma volume or activating the RAAS and SNS. Therefore, this combination therapy may be a promising and safe strategy for long-term cardiorenal protection, particularly in patients with fluid retention.

## Figures and Tables

**Figure 1 diagnostics-14-01194-f001:**
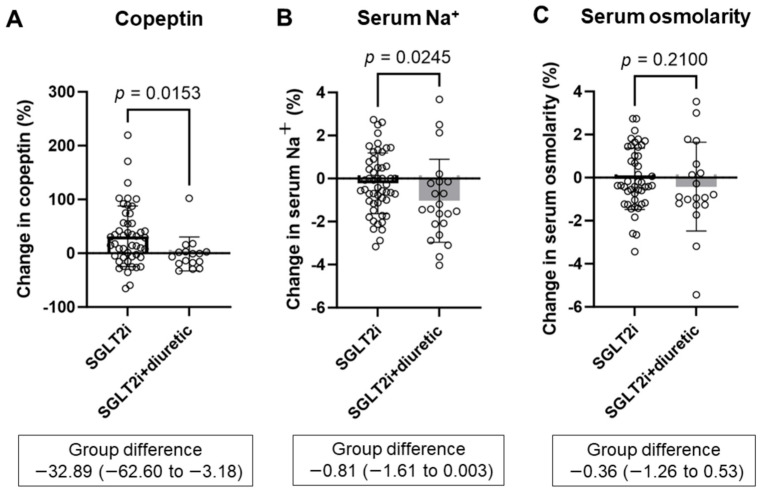
Comparing percentage changes in copeptin (**A**), serum Na^+^ (**B**), and serum osmolarity (**C**) between the SGLT2i group and the SGLT2i + diuretic groups after 6 months. SGLT2i, SGLT2 inhibitor dapagliflozin; SGLT2i + diuretic, dapagliflozin, and diuretic (loop or thiazide diuretic).

**Figure 2 diagnostics-14-01194-f002:**
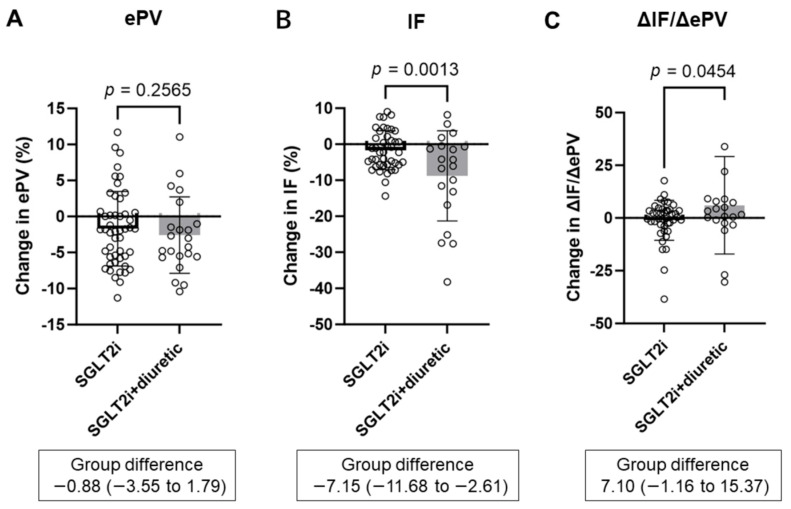
Comparing percentage changes in estimated plasma volume (ePV) (**A**), interstitial fluid (IF) (**B**), and ΔIF-to-ΔePV ratio (ΔIF/ΔePV) (**C**) between the SGLT2i and SGLT2i + diuretic groups after 6 months. SGLT2i, SGLT2 inhibitor dapagliflozin; SGLT2i + diuretic, dapagliflozin, and diuretic (loop or thiazide diuretic).

**Figure 3 diagnostics-14-01194-f003:**
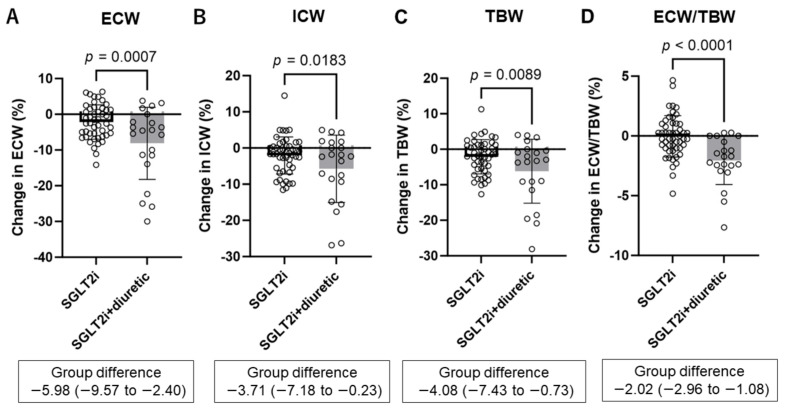
Comparing percentage changes in extracellular water (ECW) (**A**), intracellular water (ICW) (**B**), total body water (TBW) (**C**), and ECW-to-TBW ratio (ECW/TBW) (**D**) between the SGLT2i and SGLT2i + diuretic groups after 6 months. SGLT2i, SGLT2 inhibitor dapagliflozin; SGLT2i + diuretic, dapagliflozin, and diuretic (loop or thiazide diuretic).

**Figure 4 diagnostics-14-01194-f004:**
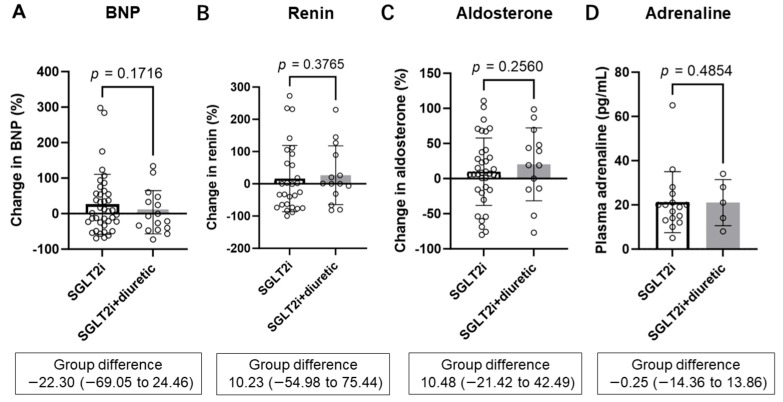
Comparing percentage changes in brain natriuretic peptide (BNP) (**A**), serum renin (**B**), and serum aldosterone (**C**) between the SGLT2i and SGLT2i + diuretic groups after 6 months. Comparing plasma adrenaline between the SGLT2i group and the SGLT2i + diuretic groups at 6 months (**D**) SGLT2i, SGLT2 inhibitor dapagliflozin; SGLT2i + diuretic, dapagliflozin, and diuretic (loop or thiazide diuretic).

**Figure 5 diagnostics-14-01194-f005:**
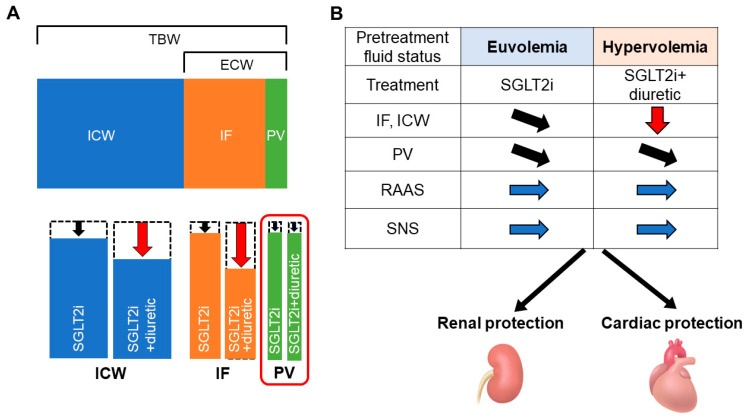
(**A**) Change in body fluid volume in the SGLT2 inhibitor-treated patients with and without diuretics. After 6 months of treatment, intracellular water (ICW), interstitial fluid (IF), extracellular water (ECW), and total body water (TBW) were reduced to a greater degree in the SGLT2i + diuretic group than that in the SGLT2i group, although the reduction in circulating plasma volume (PV) was not different between the groups. (**B**) Proposal treatment option based on pretreatment fluid status. The avoidance of excessive PV reduction or activating the renin–angiotensin–aldosterone system (RAAS) and sympathetic nervous system (SNS) both in SGLT2i and SGLT2i + diuretic groups may contribute to long-term cardiorenal protection.

**Table 1 diagnostics-14-01194-t001:** Baseline characteristics of SGLT-2 inhibitor-treated patients with and without diuretics.

Characteristics	SGLT2i (*n* = 53)	SGLT2i + Diuretic (*n* = 23)	*p*-Value
Age, years	67.0 [57.0–70.5]	73.0 [68.0–78.0]	0.006
Male gender, *n* (%)	32 (60.4)	16 (69.6)	0.442
Body weight, kg	70.5 [60.3–77.6]	67.2 [62.5–73.8]	0.289
BMI, kg/m^2^	27.1 (4.5)	27.0 (5.0)	0.452
Diabetes, *n* (%)	28 (52.8)	15 (68.2)	0.217
Systolic BP, mmHg	136 (19)	137 (20)	0.393
Diastolic BP, mmHg	75 [69–84]	70 [63–79]	0.092
Heart rate, beats/min	79 (12)	75 (12)	0.163
Hemoglobin, g/dL	13.1 (1.8)	11.5 (1.9)	0.001
Hematocrit, %	38.5 [35.5–43.1]	34.3 [31.1–38.5]	0.001
Plasma glucose, mg/dL	134 [114–160]	156 [124–184]	0.019
HbA1c, %	7.2 (0.8)	7.4 (1.2)	0.280
BNP, pg/mL	19.2 [8.5–34.9]	78.3 [28.6–354.0]	0.017
Serum albumin, g/dL	4.1 [3.8–4.3]	3.8 [3.1–4.1]	0.004
BUN, mg/dL	24.9 [16.6–33.4]	33.5 [19.4–58.3]	0.002
Serum creatinine, mg/dL	1.51 [0.98–2.18]	1.89 [1.41–2.36]	0.103
eGFR, mL/min/1.73 m^2^	34.0 [21.8–51.1]	23.3 [19.5–39.2]	0.031
Uric acid, mg/dL	5.9 (1.3)	6.2 (1.2)	0.174
Serum Na^+^, mEq/L	141 [139–142]	141 [139–142]	0.338
Serum K^+^, mEq/L	4.5 (0.5)	4.6 (0.5)	0.329
Copeptin, pmol/L	12.5 [7.1–30.4]	27.9 [14.2–65.6]	0.012
Renin, pg/mL	29.9 [8.5–82.2]	62.4 [27.1–113.6]	0.043
Aldosterone, pg/mL	325 (188)	331 (188)	0.468
Proteinuria, g/gCr	1.07 [0.30–3.54]	0.57 [0.18–2.24]	0.294
ICW, L	21.10 [18.00–24.10]	19.30 [17.60–22.90]	0.251
ECW, L	13.62 (3.30)	14.17 (2.53)	0.245
TBW, L	33.90 [29.80–39.30]	33.30 [29.85–40.00]	0.445
ECW/TBW	0.392 [0.387–0.400]	0.409 [0.402–0.417]	<0.0001
Fat mass, kg	22.0 [16.5–28.3]	22.3 [15.6–26.7]	0.461
Bone mineral content, kg	2.5 [2.1–2.8]	2.4 [2.0–2.7]	0.189
Concomitant diuretics, %	0	100	<0.0001
Loop diuretics (furosemide, torsemide, or azosemide), %	0.0	87.0	<0.0001
Thiazide diuretics (trichlormethiazide or hydrochlorothiazide), %	0.0	17.4	0.002
MR antagonist, %	3.8	17.4	0.055
Tolvaptan, %	1.9	13.0	0.058
ACE inhibitor or ARB, %	62.3	59.1	0.798
Calcium channel blocker, %	67.9	90.9	0.026

Data are presented as mean (standard deviation) or median [interquartile range], as appropriate. BMI, body mass index; BP, blood pressure; BNP, brain natriuretic peptide; BUN, blood urea nitrogen; eGFR, estimated glomerular filtration rate; ICW, intracellular water; ECW, extracellular water; TBW, total body water; MR, mineralocorticoid receptor; ACE, angiotensin-converting enzyme; ARB, angiotensin receptor blocker.

**Table 2 diagnostics-14-01194-t002:** Change in fluid volume status in the SGLT2 inhibitor-treated patients with and without diuretic.

	SGLT2i (*n* = 53)			SGLT2i + Diuretic (*n* = 23)			Comparison of Changes
	Baseline Mean (SD)	6 Months Mean (SD)	Change [95% CI]	Baseline Mean (SD)	6 Months Mean (SD)	Change [95% CI]	*p* Value
ePV, L	2.66 (0.53)	2.51 (0.06)	−0.10 [−0.15 to −0.06]	2.75 (0.46)	2.58 (0.41)	−0.16 [−0.24 to −0.08]	0.0908
IF, L	11.22 (2.83)	10.83 (2.90)	−0.19 [−0.37 to −0.06]	11.42 (2.26)	10.62 (2.38)	−1.02 [−1.67 to −0.37]	0.0007
ECW, L	13.62 (3.30)	13.27 (3.12)	−0.23 [−0.48 to 0.03]	14.17 (2.53)	13.21 (2.57)	−1.16 [−1.80 to −0.53]	0.0006
ICW, L	21.15 (5.06)	20.62 (5.01)	−0.39 [−0.74 to −0.05]	20.33 (3.93)	19.30 (4.67)	−1.03 [−1.75 to −0.32]	0.0332
TBW, L	34.77 (8.33)	33.89 (8.09)	−0.62 [−1.19 to −0.05]	34.49 (6.34)	32.90 (6.93)	−2.05 [−3.35 to −0.74]	0.0096
ECW/TBW	0.393 (0.010)	0.393 (0.002)	0.001 [−0.002 to 0.004]	0.411 (0.018)	0.403 (0.003)	−0.009 [−0.013 to −0.005]	0.0003

A *t*-test was performed on the change between the SGLT-2i group and SGLT-2i+diuretic group and a *p*-value was caliculated. Data are presented as mean (standard deviation). ePV, estimated plasma volume; ECW, extracellular water; TBW, total body water; ICW, intracellular water; IF, interstitial fluid. SGLT2i, SGLT2 inhibitor dapagliflozin; SGLT2i + diuretic, dapagliflozin and diuretics (loop or thiazide diuretic); SD, standard deviation; 95% CI, 95% confidence interval.

## Data Availability

The raw data supporting the conclusion of this article will be made available by the authors without reservation.

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
