# Peer review of "Sodium–Glucose Cotransporter 2 Inhibitor Combined with Conventional Diuretics Ameliorate Body Fluid Retention without Excessive Plasma Volume Reduction"

_diagnostics, 2024, doi:10.3390/diagnostics14111194_

Round 1

Reviewer 1 Report

Comments and Suggestions for Authors

The research article entitled "Copeptin and body fluid responses to the combined use of SGLT2 inhibitor and diuretics in patients with CKD: the DAPA-BODY trial" submitted by Asakura-Kinoshita and colleagues addresses an important and clinically relevant question regarding the feasibility of SGLT2 inhibitors in as cardiorenal protectant in diabetic patients.

My specific comments about the manuscript are as under:

1.    The title needs to be more specific and should mention the findings.

2.    The introduction is lengthy while missing key details important for the current study e.g. As Copeptin is not a conventional marker for plasma vasopressin levels, the authors should at least dedicate a few lines on the diagnostic reliability of copeptin as the current study heavily relies on copeptin levels. While the same holds true for the abstract section.

3.    The authors should also dedicate a few lines to explain what the authors meant by conventional diuretics.

4.    The authors should also emphasize the reasons for the specific selection of dapagliflozin out of multiple gliflozin members available in the introduction.

5.    The specific loop and thiazide diuretics used in the study should be named in the materials and methods section.

6.    Section 2.2 Study endpoints: Only the abbreviation TBW was used and total body water is missing. TBW is the abbreviation.

7.    Table 1. Instead of only mentioning the terms loop diuretics and thiazide diuretics, the authors should also include the specific names of the drugs used.

8.    Figure 1, 2, 3, 4. The graphs look hazy and dull with low resolution. Seems like the graphs were copied and pasted as jpeg which generally results in the deterioration of the resolution. The authors must replace these graph figures with clear high-resolution graphs figures.

9.     Figure 1, 2, 3, 4. These figures represent percentage changes. The two independent values of different parameters taken at baseline and after 6 months for both treatments (SGLT2i alone + SGLT2i combination) must be presented in the manuscript as a graph figure.

10.  The second table has no title number or description.

Author Response

Comments and Suggestions for Authors

The research article entitled "Copeptin and body fluid responses to the combined use of SGLT2 inhibitor and diuretics in patients with CKD: the DAPA-BODY trial" submitted by Asakura-Kinoshita and colleagues addresses an important and clinically relevant question regarding the feasibility of SGLT2 inhibitors in as cardiorenal protectant in diabetic patients.

Thank you for reviewing our manuscript and providing comments to improve it. We have revised the manuscript based on your comments and our responses are as follows:

My specific comments about the manuscript are as under:

  1. The title needs to be more specific and should mention the findings.

Response: The title was updated to include the major findings of this study.

  1. The introduction is lengthy while missing key details important for the current study e.g. As Copeptin is not a conventional marker for plasma vasopressin levels, the authors should at least dedicate a few lines on the diagnostic reliability of copeptin as the current study heavily relies on copeptin levels. While the same holds true for the abstract section.

Response: As the reviewer suggested, we added a few lines on the diagnostic reliability of copeptin in the introduction section (page 2-3) and the abstract.

  1. The authors should also dedicate a few lines to explain what the authors meant by conventional diuretics.

Response: We added a detailed explanation of “conventional diuretics” in the abstract, introduction (page 3), materials and methods (page 3), and Table 1.

  1. The authors should also emphasize the reasons for the specific selection of dapagliflozin out of multiple gliflozin members available in the introduction.

Response: We emphasized the reasons for selecting dapagliflozin in the introduction section (page 3).

  1. The specific loop and thiazide diuretics used in the study should be named in the materials and methods section.

Response: We added the specific names of loop and thiazide diuretics in the materials and methods section (page 3).

  1. Section 2.2 Study endpoints: Only the abbreviation TBW was used and total body water is missing. TBW is the abbreviation.

Response: We changed accordingly (page 4).

  1. Table 1. Instead of only mentioning the terms loop diuretics and thiazide diuretics, the authors should also include the specific names of the drugs used.

Response: We added the specific names of the diuretics in Table 1.

  1. Figure 1, 2, 3, 4. The graphs look hazy and dull with low resolution. Seems like the graphs were copied and pasted as jpeg which generally results in the deterioration of the resolution. The authors must replace these graph figures with clear high-resolution graphs figures.

Response: We replaced previous figures with high-resolution figures.

  1. Figure 1, 2, 3, 4. These figures represent percentage changes. The two independent values of different parameters taken at baseline and after 6 months for both treatments (SGLT2i alone + SGLT2i combination) must be presented in the manuscript as a graph figure.

Response: We added graph figures accordingly (Supplementary Figures 1-4).

  1. The second table has no title number or description.

Response: We confirmed the title number and description of Table 2.

Reviewer 2 Report

Comments and Suggestions for Authors

Dear Authors,

my concerns are as follows:

Technical errors

On page 3 it is stated that "this is a randomized study", however, on page 12 in the limitations the same is  mentioned as "not a randomized study".

Several primary endpoints are specified, BUT there should be only one. The Endpoints are not described correctly.

I consider it unnecessary to describe results that are not significant, let alone  illustrate them in figures.

 Figure 5  illustrates the relationship between PV, IF, ICW in a strange manner. It looks like the PV is in the head, the IF is in the body, and the ICW is in the legs.

Not all data are presented  in the Statistical analysis section.

Errors in Methods

I consider the calculation of fluid status variables for the studied category of patients using formulas to be incorrect; in this case, objective assessment methods are needed.

Methodological errors

The groups are heterogeneous in terms of baseline characteristics (Table 1), which are important for this study and can significantly affect the results (Age, Hb, Ht, glucose, albumin, BYN, eGFR, ECW/TBW, use of diuretics). The study used different diuretics.

Author Response

Dear Authors,

my concerns are as follows:

Thank you for reviewing our manuscript and providing comments to improve it. We have revised the manuscript based on your comments and our responses are as follows:

Technical errors

On page 3 it is stated that "this is a randomized study", however, on page 12 in the limitations the same is mentioned as "not a randomized study".

Response: This study is non-randomized. We corrected the wrong description on page 3.

Several primary endpoints are specified, BUT there should be only one. The Endpoints are not described correctly.

Response: We changed the primary endpoint definition accordingly (page 4).

I consider it unnecessary to describe results that are not significant, let alone illustrate them in figures.

Response: In this study, the lack of significant differences in several parameters such as plasma volume, RAAS, and SNS is important and similar to those parameters that are significantly different. Therefore, we described results that are not significant in the results section (pages 6-9).

 Figure 5  illustrates the relationship between PV, IF, ICW in a strange manner. It looks like the PV is in the head, the IF is in the body, and the ICW is in the legs.

Response: We changed the illustration of the body fluid compartment. Furthermore, we added a new illustration summarizing the current data and possible cardiorenal benefits.

Not all data are presented in the Statistical analysis section.

Response: We confirmed and changed the description of the Statistical analysis section.

Errors in Methods

I consider the calculation of fluid status variables for the studied category of patients using formulas to be incorrect; in this case, objective assessment methods are needed.

Response: We agree with the reviewer's concern. Because plasma volume was estimated using the formulas, we added the concern for the accuracy of fluid status assessment as a limitation (page 13).

Methodological errors

The groups are heterogeneous in terms of baseline characteristics (Table 1), which are important for this study and can significantly affect the results (Age, Hb, Ht, glucose, albumin, BYN, eGFR, ECW/TBW, use of diuretics). The study used different diuretics.

Response: We agree with the reviewer's point. We added the possibility that several different baseline characteristics and diuretics may affect the results of this study (page 13).

Reviewer 3 Report

Comments and Suggestions for Authors

The present manuscript investigates the effect of SGLT2 inhibitor  treatment with-versus  without diuretics on body volume status in ckd patients, in a non-randomized manner.

SGLT2 inhibitors have become milestone drugs in the treatment of diabetes mellitus, heart insufficiency  and chronic kidney diseases. The interaction and/or synergism with eg diuretics in clinical settings is of importance  and respective data about  fluid and water balance are important. The present study adds to a better understanding of the complex network of fluid(water)-volume regulation in so treated patients.

Comments:

-The various beneficial effects of SGLT2 inhibitors have been demonstrated in conjunction with ACEi or ARBs ( partially linked via TG -feedback).  In the CREDENCE  or EMPA-KIDNEY studies patients had these co-medications in over 80 %. In the present study this was only the case in about 60 %. The authors should comment on this in their discussion section with regard to their results.

-There might be a difference in co-administration of either a loop diuretic or a thiazide, since the latter one might increase aquaporin expression in the collecting tubule.

-The co-administration of a MR antagonist , as this was the case in about 17 % of cases  in the diuretic arm of  the present study, also has antidiuretic properties. 

The authors should discuss these points in the manuscript.

-Although not tested so far in greater studies, the authors should speculate about reasonable diuretic dosages in these patients on the long run.

-The effect of SGLT2 inhibitors in the kidney also depends on the degree of renal insufficiency and is therefore reduced with failing kidney function. In this study egfr ranged roughly between 20 and 50 ml/min/1.73m2. How this impacts on the data obtained?

Author Response

Comments and Suggestions for Authors

The present manuscript investigates the effect of SGLT2 inhibitor treatment with-versus  without diuretics on body volume status in ckd patients, in a non-randomized manner.

SGLT2 inhibitors have become milestone drugs in the treatment of diabetes mellitus, heart insufficiency and chronic kidney diseases. The interaction and/or synergism with eg diuretics in clinical settings is of importance and respective data about fluid and water balance are important. The present study adds to a better understanding of the complex network of fluid(water)-volume regulation in so treated patients.

Thank you for reviewing our manuscript and providing comments to improve it. We have revised the manuscript based on your comments and our responses are as follows:

Comments:

-The various beneficial effects of SGLT2 inhibitors have been demonstrated in conjunction with ACEi or ARBs (partially linked via TG -feedback). In the CREDENCE or EMPA-KIDNEY studies patients had these co-medications in over 80 %. In the present study this was only the case in about 60 %. The authors should comment on this in their discussion section with regard to their results.

Response: We think that the relatively low usage rate of ACEi or ARBs had a weak effect on body fluid status, a primary result of this study. This is because 1) the primary efficacy of ACEi or ARBs is not diuretic action and 2) the usage rate of ACEi or ARBs was similar between the two groups in this study. We added these descriptions in the discussion section (page 13).

-There might be a difference in co-administration of either a loop diuretic or a thiazide, since the latter one might increase aquaporin expression in the collecting tubule.

Response: We think this is an important point. However, unfortunately, this study includes only two patients of thiazide without a loop diuretic (the other two patients were the combined use of loop and thiazide diuretics). Therefore, we could not perform a statistical comparison between the two drugs. We added this point and the need for further studies in the discussion section (pages 13).  

-The co-administration of a MR antagonist, as this was the case in about 17 % of cases in the diuretic arm of the present study, also has antidiuretic properties.

The authors should discuss these points in the manuscript.

Response: This is an important point. Similar to a thiazide, this study includes only two patients of MR antagonist without a loop diuretic (the other two patients were the combined use of loop diuretic and MR antagonist). We added this description and the need for further requirements in the discussion section (pages 13).  

-Although not tested so far in greater studies, the authors should speculate about reasonable diuretic dosages in these patients on the long run.

Response: The higher doses of diuretics may accelerate hypovolemia and the activation of the renin-angiotensin-aldosterone system and sympathetic nervous system. Therefore, lower doses of diuretics (less than 40 mg of furosemide equivalent) for long-term cardiorenal protection. may be appropriate to the extent that fluid retention is controlled. We added this point in the discussion section (page 13).

-The effect of SGLT2 inhibitors in the kidney also depends on the degree of renal insufficiency and is therefore reduced with failing kidney function. In this study egfr ranged roughly between 20 and 50 ml/min/1.73m2. How this impacts on the data obtained?  

Response: According to the reviewer's comment, we analyzed the correlation between baseline eGFR and changes in body fluid parameters, which all had no significant correlation (page 9). Therefore, different baseline renal function in enrolled patients might not strongly affect the data obtained. We added these descriptions in the discussion section (page 12).

Round 2

Reviewer 2 Report

Comments and Suggestions for Authors

Dear Author,

the revised version seems quite satisfactory.

Reviewer 3 Report

Comments and Suggestions for Authors

The questions raised have been adequately answered and discussed in the manuscript